# Pheromone Guidance of Polarity Site Movement in Yeast

**DOI:** 10.3390/biom12040502

**Published:** 2022-03-26

**Authors:** Katherine C. Jacobs, Daniel J. Lew

**Affiliations:** Department of Pharmacology and Cancer Biology, Duke University, Durham, NC 27708, USA; katherine.jacobs@duke.edu

**Keywords:** cell polarity, mating, pheromone, yeast

## Abstract

Cells’ ability to track chemical gradients is integral to many biological phenomena, including fertilization, development, accessing nutrients, and combating infection. Mating of the yeast *Saccharomyces cerevisiae* provides a tractable model to understand how cells interpret the spatial information in chemical gradients. Mating yeast of the two different mating types secrete distinct peptide pheromones, called **a**-factor and α-factor, to communicate with potential partners. Spatial gradients of pheromones are decoded to guide mobile polarity sites so that polarity sites in mating partners align towards each other, as a prerequisite for cell-cell fusion and zygote formation. In ascomycetes including *S. cerevisiae*, one pheromone is prenylated (**a**-factor) while the other is not (α-factor). The difference in physical properties between the pheromones, combined with associated differences in mechanisms of secretion and extracellular pheromone metabolism, suggested that the pheromones might differ in the spatial information that they convey to potential mating partners. However, as mating appears to be isogamous in this species, it is not clear why any such signaling difference would be advantageous. Here we report assays that directly track movement of the polarity site in each partner as a way to understand the spatial information conveyed by each pheromone. Our findings suggest that both pheromones convey very similar information. We speculate that the different pheromones were advantageous in ancestral species with asymmetric mating systems and may represent an evolutionary vestige in yeasts that mate isogamously.

## 1. Introduction

Many cells decode spatial gradients of chemical signals to navigate through their environment, including immune cells migrating to wound sites, gametes locating each other for fertilization, and axons growing toward synaptic targets [1,2,3]. Our understanding of the molecular mechanisms that allow cells to decode the spatial information in chemical gradients is incomplete. Mating cells of the budding yeast *Saccharomyces cerevisiae* use gradients of pheromones to locate their partners, providing a tractable model in which to study gradient decoding. 

Haploid yeast cells of opposite mating type, MAT**a** or MATα, can mate to form a diploid zygote. Cells of each mating type secrete a unique peptide pheromone and express a G-protein coupled receptor on the cell surface that recognizes the pheromone secreted by cells of the opposite mating type (Figure 1A). Pheromone binding to the receptor triggers activation of a MAPK signaling cascade, which results in expression of mating-specific genes, cell-cycle arrest in G1 phase, and polarization of the conserved Rho-family GTPase Cdc42 [4]. Active Cdc42 becomes concentrated at a cortical “polarity site”, to which it recruits effectors (proteins that specifically bind active Cdc42) that orient actin cables and promote local vesicle fusion with the plasma membrane [5]. Vesicles deliver cell-wall remodeling enzymes that allow the mating partners to grow toward each other and degrade the intervening cell walls prior to membrane fusion [6]. For this program to succeed, a cell’s polarity site must be oriented toward its partner. 

Correct orientation of polarity requires reciprocal orientation of the partner’s polarity site: a cell cannot stably orient toward a partner that lacks a polarity site or has a mis-oriented polarity site [7]. This reciprocity requirement implies that partners of both mating types must decode pheromone gradients to orient polarity, consistent with an isogamous mating process where each partner conveys similar information to the other. However, the pheromones secreted by cells of different mating types have distinct physical properties that seem likely to affect their spatial concentration profiles and, therefore, the spatial information provided to the receiving cell. 

Mature α-factor is a hydrophilic 13-residue peptide with no post-translational modifications, while mature **a**-factor is a 12-residue peptide that is hydrophobic as a result of farnesylation and carboxyl methylation [8] (Figure 1A). α-factor is secreted via exocytosis and given its small size it presumably diffuses rapidly in aqueous environments, leading to the expectation that there would be a steep gradient of α-factor decaying from the secretion site. **a**-factor is secreted by a transmembrane transporter (Ste6) but given its hydrophobic character its subsequent fate is less obvious. One could imagine that secreted **a**-factor may first accumulate on the outer leaflet of the secreting cell’s plasma membrane, before being gradually shed; or perhaps it might attach to carrier proteins, or to vesicles shed from the secreting cell, sharing their much slower mobility [9]. It might also form micelles, or accumulate at the interfaces between aqueous media and nearby surfaces, or at nearby air-water interfaces. All of these circumstances seem very likely to impact the spatial concentration profile of **a**-factor pheromone.

Spatial concentration profiles may also be impacted by other factors, and particularly by proteases. MAT**a** cells secrete a protease, Bar1, that specifically degrades α-factor [10,11] (Figure 1A). Bar1 has been shown to affect α-factor concentration profiles both in mating mixes [12,13] and in synthetic pheromone gradients created by microfluidic devices [14,15]. Although an activity capable of degrading **a**-factor has also been reported [16], its molecular nature remains unknown. A secreted protein called Afb1 (**a**-factor blocker) was identified in an ingenious series of experiments using “transvestite” yeast cells, and can act extracellularly to reduce **a**-factor action, perhaps via sequestration [17]. However, mutants lacking Afb1 mated as well as wild-type cells in lab conditions, and it is unclear whether and how **a**-factor profiles might be affected by secreted factors. 

If the spatial distributions of α-factor and **a**-factor differ due to their physical properties, or mechanisms of secretion, or unique extracellular modulators, then the two pheromones could provide quite different information to the receiving cells [8,17]. Indeed, a recent study proposed that α-factor provides a non-directional long-range signal to alert distant MAT**a** cells that they should prepare for mating, while **a**-factor provides a directional but short-range signal that influences polarity in the receiving MATα cells [18]. A similar proposal emerged from studies with the fission yeast *Schizosaccharomyces pombe*, which also uses one unprenylated pheromone and one prenylated pheromone [19]. Without a way to directly compare **a**-factor and α-factor concentration profiles around mating cells, we can only infer the spatial information conveyed by each profile from examination of how cells of different mating types respond to their partners. 

The spatial information provided by pheromone gradients is translated into orientation of each partner’s polarity site towards the other. Polarity sites co-concentrate active Cdc42 along with its regulators and effectors at a region of the plasma membrane [5]. Imaging of mating cells revealed an initial “indecisive” period during which polarity factors behave erratically, forming clusters at various locations that can appear, move, and disappear in an apparent search process [20]. After a variable interval, each cell develops a strong and stable polarity site oriented towards the partner, a process termed “commitment” [7,13,20]. Presumably, spatial information is conveyed during the indecisive period, but it is challenging to track the erratic polarity factor distribution during that stage. In contrast, cells genetically engineered to activate the mating MAPK, Fus3, display a search process in which a single strong polarity site moves around the cortex in a manner that can be tracked using a custom image analysis pipeline [21,22]. This provides an opportunity to quantitatively compare the polarity-site guidance provided by the different pheromones.

In this study, we investigated the spatial information that α-factor and **a**-factor provide to mating partners. We show that when cells are in close proximity (touching), α-factor and **a**-factor provide indistinguishable guidance for polarity site movement to cells of opposite mating type, despite their different physical properties. Such guidance appears optimized for isogamous mating but raises the question of why the pheromones differ in the first place. We speculate that different pheromones may be a vestige retained from ancestors with anisogamous mating systems, or that they may confer some advantage in different mating environments.

## 2. Materials and Methods

### 2.1. Yeast Strains

Yeast strains were constructed with standard molecular genetic techniques and are listed in Table 1. All strains were generated in the YEF473 background (*his3-*Δ*200 leu2-*Δ*1 lys2-801 trp1-*Δ*63 ura3-52*). The following alleles were previously described: *SPA2-mCherry:hyg^R^* [23], *rsr1::HIS3* [24], *WHI5-GFP:HIS5* [7], *cdc24-m1:TRP1* [21], *SPA2-GFP:HIS3* and *ste5:P_GAL1_-STE5-CTM:P_ADH1_-GAL4BD-hER-VP16:LEU2* [20]. The latter replaces the endogenous *STE5* gene with two genes: one encodes a hybrid transcription factor that promotes expression from the *GAL1* promoter only in the presence of β-estradiol, and the second drives expression of the membrane-targeted Ste5-CTM from the *GAL1* promoter. As a result, our strains are sterile (due to a lack of Ste5) in the absence of β-estradiol and are induced to activate the mating MAPK pathway in the presence of β-estradiol.

To delete *STE2*, *STE3* and *GPA1*, we used a PCR-based method [25] to precisely replace the open reading frame of the gene with heterologous sequences. Briefly, we used primers with 50 bp of homology to the 5′ and 3′ untranslated regions of the gene to be deleted to amplify a selectable marker from a template, transformed the PCR product into yeast, selected for the marker, and checked correct integration by PCR using flanking DNA primers. The templates were pFA6a-kanMX6 [26] for *GPA1* and pRS400-kanMX4 [27,28] for *STE2* and *STE3*. 

### 2.2. Live Cell Microscopy

Cells were grown overnight at 30 °C in complete synthetic medium (CSM, MP Biomedicals, Solon, OH, USA) with 2% dextrose to mid-log phase (10^6^–10^7^ cells/mL). Cells were pre-treated with 20 nM β-estradiol (Sigma-Aldrich, St. Louis, MO, USA) for 3 h to induce expression of Ste5-CTM, harvested by centrifugation, mixed with appropriate partners, mounted on CSM slabs with 2% dextrose and 20 nM β-estradiol, and sealed with petroleum jelly. This protocol causes the cells to activate the mating MAPK, arrest the cell cycle in G1, and polarize Cdc42 during the pre-treatment interval in the absence of a partner (and hence in the absence of the pheromone to which they are responsive). Mixing of pre-arrested cells of opposite mating type on the slab exposes them to pheromone and we then track the movement of the polarity site to assess the directional bias introduced by pheromone gradients. Imaging was performed at 30 °C.

Images were acquired with an Andor Revolution XD spinning disk confocal microscope (Andor Technology, Concord, MA, USA) with a CSU-X1 5000-rpm confocal scanner unit (Yokogawa, Tokyo, Japan) and a UPLSAPO 100x/1.4 oil-immersion objective (Olympus, Tokyo, Japan), controlled by MetaMorph software (Molecular Devices, San Jose, CA, USA). Images were captured by an iXon 897 EMCCD camera (Andor Technology) or an iXon Life 888 EMCCD camera (Andor Technology). 

Z-stacks with 15 z-steps of 0.48 μm were acquired at 2 min intervals. Laser power was set to 10% of maximum for 488 nm and 13% of maximum for 561 nm. Exposure time was 250 ms and EM gain was 200.

### 2.3. Polarity Site Tracking

Polarity site tracking was performed as in [21] using Volocity software (Quorum Technologies Inc., Puslinch, ON, Canada). The 3D centroid of each polarity site was located by thresholding the Spa2-mCherry signal. Centroids were tracked until the Spa2-mCherrry signal overlapped with the Spa2-mCherry signal of the partner’s polarity site. Fluorescent 0.2 µm TetraSpeck beads (Thermo Fisher Scientific, Waltham, MA, USA) were added to the slab to account for stage drift; the centroid of the bead was used as an origin to track the centroid of the polarity site at each timepoint. Analysis excluded timepoints where no polarity site was detected or where multiple polarity sites were detected. 

### 2.4. Determination of Polarity Site Movement Direction

Determination of polarity site movement direction was performed as in [21]. The polarity site can only move along the cell cortex, so we developed a method to determine the optimal direction of polarity site movement within that constraint. For a given step, we used the current location of the polarity site and the locations of the site at two surrounding timepoints to define a plane roughly tangential to the surface of the cell. We projected the polarity site of the mating partner onto that plane. The vector connecting the polarity site to the site of the mating partner defines the optimal direction. The angle theta (θ) is the angle between the vector defining the optimal direction and the actual step of the polarity site. Angles were determined for all timepoints until the polarity sites of mating partners encounter each other: tracking direction after that point was no longer possible.

### 2.5. Scoring Polarity Site Encounters

Polarity sites were analyzed using maximum projections of z-stacks in ImageJ (National Institutes of Health, Bethesda, MD, USA). An encounter was recorded when the polarity sites (visualized using Spa2 probes) in partner cells approached within 1 µm of each other. Images were visually inspected to identify times with potential encounters. A line 10 pixels wide was drawn across the two polarity sites to generate a line scan of fluorescence intensity. Each polarity site appeared as a peak of signal intensity on the line scan and the distance between the two peaks was measured. An encounter was recorded if peaks were less than 1 µm apart.

### 2.6. Statistical Analysis

Two-sample Kolmogorov-Smirnov (KS) tests were performed in Microsoft Excel using the Real Statistics Resource Pack add-in (Release 5.11) developed by Charles Zaiontz.

## 3. Results

### 3.1. Spatial Information Conferred by Gradients of **a**-factor Versus α-Factor

Expression of a membrane-targeted version of the MAPK scaffold protein Ste5 (Ste5-CTM) induces strong MAPK (Fus3) activation without need for pheromone [29]. This causes haploid cells to arrest in G1 phase of the cell cycle and polarize Cdc42 [23,29]. The polarity site in such cells is mobile, and its position can be tracked using fluorescently tagged polarity markers like Spa2 (Figure 1B) [21,22,23]. In previous work, we showed that when cells of opposite mating type were separately induced to express Ste5-CTM before being mixed together, the movement of the polarity site was subsequently biased toward the mating partner: it was more likely to move toward the mating partner than away from it [21]. The strength of the directional bias was dependent on the distance between partner cells’ polarity sites: sites that were within 4 µm of each other exhibited strong directional bias while sites farther apart did not. However, that study did not address whether the directional bias reflected primarily a bias of the MAT**a** cell’s polarity site towards the MATα cell, or a bias of the MATα cell’s polarity site towards the MAT**a** cell, or both.

In most lab mating assays, cells are mixed at high density so that a given cell may have more than one potential partner [30,31]. The presence of more than one partner makes it hard to identify which partner may be biasing polarity site movement. Thus, to unambiguously address how polarity site movement is oriented with respect to a partner, we initially restricted our analysis to isolated pairs of cells in which each cell had only one potential (touching) mating partner. As pheromone is emitted from the polarity site [7], we measured the angle between the direction of polarity site movement and an “optimal” direction towards the polarity site of the mating partner (see Methods). The direction of movement was determined in each 2 min imaging interval, and the distribution of angles is shown in Figure 1B. Our analysis showed that polarity sites in MAT**a** and MATα cells exhibited very similar directional bias towards each other (Figure 1B). This finding implies that under these mating conditions, α-factor and **a**-factor provide similar directional information to the receiving cell.

Given the differences between the two pheromones noted in the introduction, it was quite surprising that each appeared to provide similar spatial information, and we wondered whether a similar conclusion would hold in more crowded conditions that mimic more physiological environments [32]. In more crowded conditions, pheromone gradients may be more confusing and less informative, lengthening the search process (Figure 2A). On the other hand, it would be more likely that polarity sites might approach each other by chance, thereby shortening the search process. To assess the timing with which cells aligned their polarity sites in sparse vs. crowded conditions, we quantified how long it took for the polarity sites of partner cells to encounter each other for cells that were touching one, two, or more than two partners. Polarity site encounters occurred slightly more rapidly when cells were touching >1 potential partner (Figure 2A).

A polarity site encounter can in principle result in two possible outcomes: the polarity sites might stabilize (stop moving) and “commit” to the partner, or one or both polarity sites might keep moving (an outcome we have called “kiss-and-run”: [13]). For wild-type cells touching a single partner, most encounters resulted in commitment (Figure 2B). However, kiss-and-run encounters were more frequent when cells were touching >1 partner (Figure 2B). In summary, cells in crowded conditions show only small quantitative differences with cells in sparse conditions.

We next used the encounter assay in a separate experiment to investigate the spatial information conferred by each pheromone. To that end, we imaged mating mixes where only one partner was capable of receiving the pheromone signal: MAT**a**
*ste2*Δ cells, lacking the α-factor receptor, were mixed with wild-type MATα *STE3* cells; and wild-type MAT**a**
*STE2* cells were mixed with MATα *ste3*Δ cells, lacking the **a**-factor receptor. As before, we induced Ste5-CTM so that all cells were arrested in G1 and had mobile polarity sites, and we analyzed polarity site encounters between potential partners. 

Polarity site encounters between wild-type and receptorless cells were less frequent than encounters between wild-type cells of both mating types (Figure 3A). Presumably, only the partner with pheromone receptors is capable of biasing polarity site movement in these mixes, so encounters can only be accelerated by spatial information provided by the receptorless cell. Strikingly, encounters occurred with similar timing and frequency whether cells were able to perceive only **a**-factor (*STE3* × *ste2*Δ) or only α-factor (*STE2* × *ste3*Δ) (Figure 3A). This finding is consistent with our findings on directional bias in wild-type mixes (Figure 1B), and suggests that each pheromone provides similar information to the partner that perceives it. 

When one partner lacked its pheromone receptors, we observed only “kiss-and-run” encounters (Figure 3B). Thus, pheromone signaling in both partners is critical for commitment: without sensing pheromone there is apparently no way for a cell to know that the polarity sites are aligned.

### 3.2. Mechanism of Gradient Decoding

Considerable previous work identified an elegantly simple pathway by which spatial information is conveyed from the pheromone receptor to the polarity regulator Cdc42. Pheromone-receptor binding leads to GTP-loading of Gα and release of Gβγ, which results in recruitment of the scaffold protein Far1 from the cytoplasm to the membrane site with free Gβγ; Far1 in turn recruits the Cdc42-directed guanine nucleotide exchange factor (GEF) Cdc24, which locally activates Cdc42 [33,34,35,36]. This pathway is important both for the directional bias of polarity site movement [21] and for commitment [23,37,38]. However, intriguing early reports have suggested that the pheromone receptor could provide spatial information through an additional, G-protein-independent pathway [39,40]. G-protein-independent signaling is well established in mammalian systems, where GPCRs can interact with β-arrestin as well as G-proteins [41]. However, yeast lack a clear β-arrestin homolog, and no specific G-protein-independent signaling pathway has been elucidated in the yeast pheromone response despite extensive research on this pathway.

The findings of Jackson et al. (1991) and Schrick et al. (1997) came from “partner-discrimination” assays in which MAT**a** cells are mixed with equal numbers of two different potential MATα partners: one producing wild-type levels of α-factor and the other producing no α-factor. Wild-type cells mate almost exclusively (~10^5^-fold preference) with the pheromone-producing partners, presumably because pheromone gradients allow them to be found. Cells lacking the α-factor receptor Ste2 mate with equal (low) frequency with pheromone-producing and pheromone non-producing partners. To enable receptorless mating, the mating pathway was induced genetically by either deletion of Gα (which leads to global activation of Gβγ and downstream signaling) or overexpression of Ste12 (leading to transcriptional induction of mating genes). With no Ste2 receptors, there is no way for the MAT**a** cell to tell whether a partner is producing α-factor. The surprising result was that cells that lacked subunits of the G protein but expressed Ste2 retained some (poor) ability to discriminate between pheromone-producing and pheromone non-producing partners (only 7% of matings were with non-producers) [39,40]. This would appear to indicate that Ste2 can confer some spatial information independent of the G-proteins.

To investigate whether a similar effect might be detectable in our mating conditions, we generated strains lacking Gα (*gpa1*Δ). Our strains have deleted *STE5* and can thus proliferate without Gα until expression of Ste5-CTM is induced. We imaged mating mixes of *gpa1*Δ cells induced to express Ste5-CTM and quantified how long it took polarity sites of partner cells to encounter each other. If pheromone receptors are able to provide spatial information independent of the G proteins, we would expect polarity site encounters to be more frequent in the mixes between *gpa1*Δ mutants (*gpa1*Δ × *gpa1*Δ) than in mixes between receptorless mutants (*ste2*Δ × *ste3*Δ). However, we found that encounters occurred with similar (low) frequency and timing in both mixes (Figure 4A). Thus, we find no evidence for G-protein-independent guidance of the polarity site.

To confirm that gradient decoding is mediated by Far1 downstream of the receptor and G protein, we employed a previously-identified point mutant (*cdc24-m1*) that disrupts the interaction between Far1 and Cdc24, the Cdc42-directed GEF [34,35]. Polarity site encounters in a mating mix of *cdc24-m1* mutants were comparable to both the *gpa1*Δ and receptorless mixes (Figure 4A). Thus, we cannot detect any polarity site guidance that is independent of the G protein-Far1-Cdc24 pathway in this assay.

We also imaged unilateral mating mixes where one partner had an intact Far1 pathway, and one partner was a *cdc24-m1* mutant. Polarity site encounters in these mixes occurred with similar frequency and timing to those in unilateral receptorless mating mixes (Figure 4B,C), consistent with the bilateral mixes. Moreover, polarity site encounters occurred with similar frequency regardless of which mating-type partner was mutant (Figure 4B,C), confirming that both pheromones provide similar spatial information

## 4. Discussion

In this work we examined the spatial information conveyed by each pheromone, as well as the pathway(s) that decode that information. Our findings suggest that both pheromones require a single pathway, from receptor to G protein to Far1 to Cdc24, to convey spatial guidance. Like receptorless cells, cells lacking Gα and cells with impaired Far1-Cdc24 interaction showed no detectable guidance of polarity site movement. These findings appear to contradict the conclusion from the mating partner discrimination experiments of Jackson et al. (1991) and Schrick et al. (1997), who found that receptorless mutants were more defective than mutants lacking G protein subunits. A possible avenue to reconcile these findings begins with the recognition that unlike our polarity-site-tracking experiments, the partner discrimination assay requires cell-cell fusion. This raises the possibility that the G-protein-independent role of receptors in those experiments reflects a late role in the fusion process rather than a role in guiding the polarity site towards the partner. An α-factor-dependent role of pheromone receptors at the fusion step has been previously suggested [42]. If that role is independent of G protein activation, it could explain why cells lacking G protein subunits could still enhance mating frequency with cells secreting α-factor.

Our findings suggest that despite their different physical properties, modes of secretion, and susceptibility to extracellular proteases and regulators, **a**-factor and α-factor convey very similar spatial information to guide polarity site movement in partner cells. The simplest way to explain this finding is that the spatial gradients of **a**-factor and α-factor are very similar. However, it is also possible that gradients differ in ways that are compensated for by differences in pheromone perception by MAT**a** and MATα cells. Such differences, if they exist, would have to reside in the Ste2 and Ste3 receptors themselves, as the downstream decoding pathway via G proteins, Far1, and Cdc24 appears to be identical in both cases.

The fact that pheromones with such distinct physical properties would yield very similar gradients might seem counterintuitive. If the two pheromones have different diffusion constants, are degraded at different rates, and are shed by different mechanisms, how can they yield the same spatial distributions? Below, we consider diffusion, proteolysis, and shedding in turn.

The steady state spatial distribution expected for a pheromone emitted from a point source is a concentration gradient in which concentration decays inversely with distance from the source: c(r) = S/4πDr where c(r) is the concentration at a distance r from the source, D is the diffusion constant, and S is the source strength (i.e., number of particles emitted per second) [7,43,44]. Thus, two pheromones could yield the same concentration profile despite having different diffusion constants, so long as the ratio between the source strength and the diffusion constant is the same. Additionally, even if the quantitative profiles were different, the shape of the profile would be similar, allowing a receiving cell to extract similar spatial information. For example, a weaker affinity of the receptor for one pheromone could compensate for a higher absolute concentration of that pheromone. These considerations suggest that the two pheromones could deliver similar information even if they have significantly different diffusion constants.

Interestingly, experiments in which MAT**a** cells were exposed to calibrated gradients of synthetic α-factor suggested that gradient decoding is quite robust and can occur over a wide range of mean pheromone concentrations [14,38,44,45]. Similarly, recent experiments on mating cells confronted with partners that made different amounts of pheromone suggest that cells can effectively decode pheromone gradients with different source strengths [13]. These findings raise the possibility that adequate spatial information can be conveyed from the steepness (shape) of the gradient and that cells do not rely on a narrow concentration range of pheromone.

The spatial distribution of pheromone would be impacted by degradation as well as diffusion. However, a pheromone as small as α-factor is expected to diffuse very rapidly (D~150–300 µm^2^/s) in aqueous solution, which means that it would quickly escape the locality where it was secreted. As previously noted [43], proteolysis would have to be unreasonably fast to compete with such rapid diffusional escape. In crowded conditions with limited volume, Bar1 protease becomes critical to avoid α-factor build-up leading to receptor saturation [13]. It has been speculated that local Bar1 secretion and/or retention of Bar1 in the cell wall of the secreting cell could serve to sharpen gradients of α-factor [15,43]. However, experiments in which Bar1 is provided by “helper cells” in a mating mix indicated that the cellular origin of the Bar1 does not affect its ability to promote mating [12,13]. Thus, it may be that the main role of Bar1 (and perhaps of Afb1 in the case of **a**-factor) is simply to maintain pheromone levels below saturation in confined spaces, rather than to actively shape the pheromone gradients.

Secretion of α-factor occurs through exocytosis, with each secretory vesicle estimated to deliver about 1700 molecules of α-factor [7]. In contrast, **a**-factor is pumped across the plasma membrane by the transporter Ste6, presumably one molecule at a time. Simulations indicate that vesicular release would yield large transient spikes in α-factor concentration that would dissipate rapidly (in milliseconds) after each vesicle fusion event, which is very different than the steady drip expected for **a**-factor. However, because pheromone binding/unbinding to receptors is slow, these rapid fluctuations would be averaged out and may not affect the information delivered by pheromone gradients.

A much bigger effect on pheromone gradients would result if the **a**-factor exported by Ste6 were to spread all over the emitting cell by associating with the outside of the plasma membrane. If **a**-factor were shed uniformly from the cell, rather than focally from the polarity site where Ste6 is concentrated [46], then the steepness of the **a**-factor gradient would be greatly reduced [7]. Our findings argue strongly that **a**-factor gradients are comparably steep to those of α-factor, indicating that exported **a**-factor escapes rapidly. Such escape may be facilitated by **a**-factor carrier proteins, although none have yet been identified.

If circumstances conspire to make **a**-factor and α-factor gradients comparable, then why make pheromones with different properties in the first place? The asymmetric use of prenylated and unprenylated pheromones by the two mating types is conserved among ascomycetes [8,47], which include many species that make very different male and female gametes. For example, mating in *Neurospora crassa* involves chemotropism by a large female gamete that extends a hypha towards a small and non-chemotropic male gamete [48]. In asymmetric mating scenarios, the different gametes would be expected to need different spatial guidance from pheromones, perhaps making pheromones with different properties advantageous. One possibility, then, is that these differences were selected for in ancestors with anisogamous mating, and have become vestigial in yeast cells that use isogamous mating. The fission yeast *Schizosaccharomyces pombe* is also thought to undergo isogamous mating, but recent work revealed unexpected differences between the mating types at late stages of the mating process [19,49,50]. Thus, differences between the pheromones may be important for such late events, rather than for conveying spatial information. Interestingly, basidiomycetes, most of which are thought to undergo isogamous mating, use two prenylated pheromones [51,52,53].

Another possibility is that yeast cells may also benefit from distinct pheromones for spatial guidance in specific environments. Our experiments were conducted in an aqueous environment with cells sandwiched between an agarose slab and a coverslip for ease of imaging. It remains possible that in other environments the differences between pheromones would become much more important. For example, one plausible mating scenario in the wild is in fly frass deposited on fruit surfaces [32]. This may expose cells to a waxy fruit peel or an air-water interface, which could affect the distribution of the two pheromones in distinct ways.

## Figures and Tables

**Figure 1 biomolecules-12-00502-f001:**
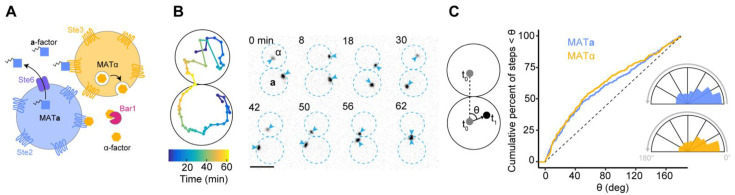
α-factor and **a**-factor provide similar spatial information. (**A**) Schematic illustrating the pheromones and receptors used by mating *S. cerevisiae* cells. MATα cells secrete the unprenylated peptide α-factor by exocytosis, while MAT**a** cells secrete the prenylated peptide **a**-factor through the exporter Ste6. α-factor is detected by the receptor Ste2 on MAT**a** cells, while **a**-factor is detected by the receptor Ste3 on MATα cells. α-factor is degraded by the protease Bar1 secreted by MAT**a** cells. (**B**) Polarity site movement during a mating event. Cells (DLY20626 and DLY20627) were treated with β-estradiol for 3 h to induce Ste5-CTM and then mixed and imaged. Inverted maximum projection images show polarity sites labeled by Spa2-mCherry (blue arrowheads). Dotted blue lines outline each cell. Time is in min. Scale bar, 5 µm. Left: Spa2 centroid trajectory (projected from 3D to 2D). (**C**) Angle of polarity site movement (θ) when partner sites are within 4 µm. θ is measured between the hypothetical optimum direction towards the partner’s polarity site (line between grey polarity site locations at the start of an imaging interval, t_0_) and the direction of actual movement of the polarity site in the next step (2 min imaging interval: black polarity site, t_1_). Cumulative distribution of angles measured at each step from MAT**a** (blue) and MATα (yellow) cells (*n* ≥ 296 steps). Inset: polar histogram of the same data. (Kolmogorov-Smirnov (KS) test = NS).

**Figure 2 biomolecules-12-00502-f002:**
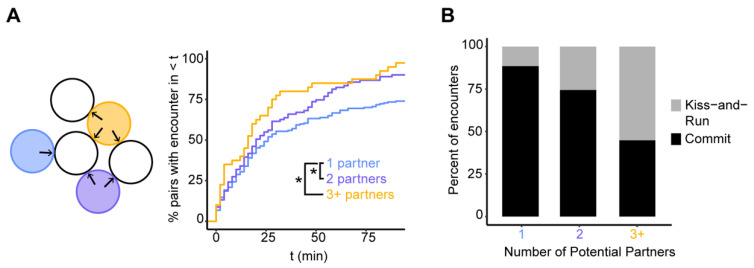
Polarity site encounters in sparse versus crowded conditions. (**A**) Left: Schematic of cells with different numbers of potential partners in a mating mixture. Potential opposite MAT partners for each colored cell are indicated by black arrows (the blue cell has one potential partner, the purple cell has two potential partners, and the yellow cell has three potential partners). Right: Cumulative distributions of the time to polarity site encounter for wild-type cells (DLY20626 and DLY20627) touching the indicated number of potential mating partners (*n* ≥ 40 pairs) (* = KS test *p* < 0.05). (**B**) Percent of encounters that result in commitment or kiss-and-run (*n* ≥ 58 encounters) for cells with the indicated number of potential mating partners.

**Figure 3 biomolecules-12-00502-f003:**
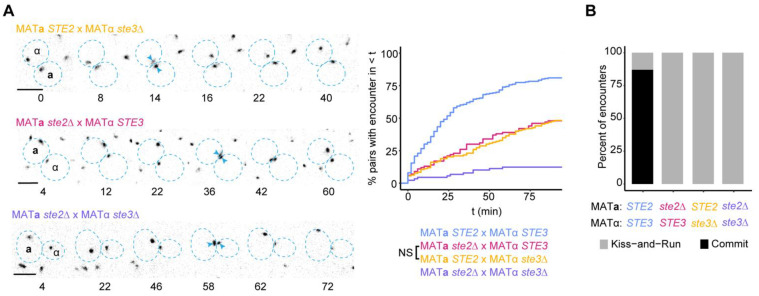
Polarity site encounters when only one partner can sense pheromone. (**A**) Left: Representative inverted maximum projection images of polarity sites labeled with Spa2-mCherry in mating mixes where one or both partners lack pheromone receptors: MAT**a**
*STE2* × MATα *ste3*Δ (DLY20626 × DLY23979), MAT**a**
*ste2*Δ × MATα *STE3* (DLY23978 × DLY20627), MAT**a**
*ste2*Δ × MATα *ste3*Δ (DLY23978 × DLY23979). Blue arrowheads indicate polarity site encounters, dotted blue lines outline each cell. Time is in min. Scale bar, 5 µm. Right: Cumulative distributions of the time to polarity site encounter in the same mating mixes (*n* ≥ 89 pairs) (NS = KS test *p* > 0.05). (**B**) Percent of encounters that result in commitment or kiss-and-run (*n* ≥ 14 encounters) from the indicated mating mixes.

**Figure 4 biomolecules-12-00502-f004:**
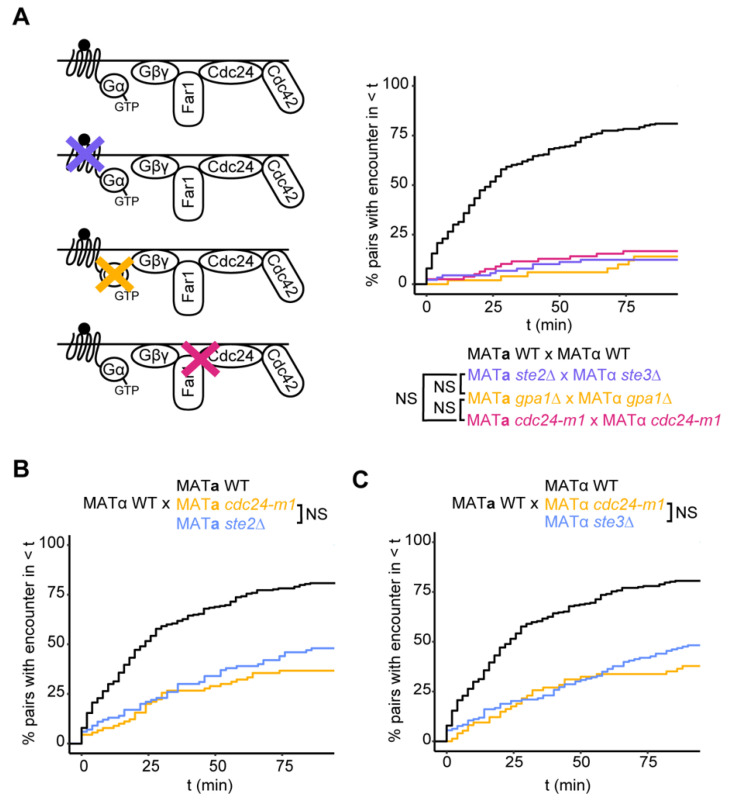
The G protein-Far1-Cdc24 pathway is required for gradient decoding. (**A**) Left: Signaling from active pheromone receptor to Cdc42, with genetic disruptions in the pathway indicated: purple ‘X’ = *ste2*Δ or *ste3*Δ, yellow ‘X’ = *gpa1*Δ, magenta ‘X’ = *cdc24-m1*. Right: Cumulative distributions of the time to polarity site encounter in mating mixes where both partners lack pheromone receptors, Gα or an intact Far1 pathway (*cdc24-m1*): MAT**a**
*ste2*Δ × MATα *ste3*Δ (DLY23978 × DLY23979), MAT**a**
*gpa1*Δ × MATα *gpa1*Δ (DLY24101 × DLY24104), MAT**a**
*cdc24-m1* × MATα *cdc24-m1* (DLY22532 × DLY22533) (*n* ≥ 50 pairs) (NS = KS test *p* > 0.05). (**B**,**C**) Cumulative distributions of the time to polarity site encounter in mating mixes where only the MATα (**B**) or MAT**a** (**C**) has an intact Far1 pathway (strains for *cdc24-m1* unilateral mixes: DLY2067 × DLY22532 or DLY20626 × 22533, strains for receptorless unilateral mixes same as in Figure 3) (*n* ≥ 74 pairs) (NS = KS test *p* > 0.05).

**Table 1 biomolecules-12-00502-t001:** Yeast strains and genotypes.

Yeast Strain	Relevant Genotype	Source
DLY20626	*MAT**a** SPA2-mCherry:hyg^R^ ste5:P_GAL1_-STE5-CTM:P_ADH1_-GAL4BD-hER-VP16:LEU2 rsr1::HIS3 WHI5-GFP:HIS5*	Ghose et al., 2021
DLY20627	*MATα SPA2-mCherry:hyg^R^ ste5:P_GAL1_-STE5-CTM:P_ADH1_-GAL4BD-hER-VP16:LEU2 rsr1::HIS3*	Ghose et al., 2021
DLY22532	*MAT**a** SPA2-mCherry:hyg^R^ ste5:P_GAL1_-STE5-CTM:P_ADH1_-GAL4BD-hER-VP16:LEU2 rsr1::HIS3 WHI5-GFP:HIS5 cdc24-m1:TRP1*	Ghose et al., 2021
DLY22533	*MATα SPA2-mCherry:hyg^R^ ste5:P_GAL1_-STE5-CTM:P_ADH1_-GAL4BD-hER-VP16:LEU2 rsr1::HIS3 cdc24-m1:TRP1*	Ghose et al., 2021
DLY23978	*MAT**a** SPA2-mCherry:hyg^R^ ste5:P_GAL1_-STE5-CTM:P_ADH1_-GAL4BD-hER-VP16:LEU2 rsr1::HIS3 WHI5-GFP:HIS5 ste2::kan^R^*	This study
DLY23979	*MATα SPA2-mCherry:hyg^R^ ste5:P_GAL1_-STE5-CTM:P_ADH1_-GAL4BD-hER-VP16:LEU2 rsr1::HIS3 ste3::kan^R^*	This study
DLY24101	*MAT**a** SPA2-GFP:HIS3 ste5:P_GAL1_-STE5-CTM:P_ADH1_-GAL4BD-hER-VP16:LEU2 rsr1::HIS3 gpa1::kan^R^*	This study
DLY24104	*MATα SPA2-mCherry:hyg^R^ ste5:P_GAL1_-STE5-CTM:P_ADH1_-GAL4BD-hER-VP16:LEU2 rsr1::HIS3 gpa1::kan^R^*	This study

## Data Availability

Experimental data used to generate figures have been deposited in GitHub: (https://github.com/kcjacobs/pheromone-guidance accessed on 6 March 2022).

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
