# Peer review of "Pheromone Guidance of Polarity Site Movement in Yeast"

_biomolecules, 2022, doi:10.3390/biom12040502_

Round 1
Reviewer 1 Report
The manuscript by Jacobs and colleagues describes the movement of polarity sites under the influence of pheromones. The author also quantitates the directional movement of polarity sites. Interestingly authors measure the polarity site encounter and suggest the possibility of commitment or kiss and run phenomena. They also test the polarity sites encounter in receptor less cells showing the information governed by each pheromone.
The overall data are convincing and well presented in the manuscript. The introduction section and result section are a bit descriptive and could be further trimmed to have focus information.
Author Response
We believe that the descriptive information is important especially for non-expert readers to follow the logic of the experiments.
Reviewer 2 Report
In this study, the authors examined the spatial information of two pheromones (a-factor and α-factor) of the budding yeast Saccharomyces cerevisiae by tracking directly their movements before cell fusion. They suggest that both pheromones provide very similar information, because the guidance of the two pheromones for cell polarity seems to be indistinguishable. Through a series of imaging experiments, the authors generate an impressive set of strains and data, supporting their findings. I believe that this study makes significant contributions to the field of the mating system in yeasts and will be of interest.
In ascomycete fungi, the pheromones for two mating types differ with respect to several properties. One pheromone is a lipid-modified peptide (hydrophobic), and is specifically secreted by the ATP-binding cassette (ABC) transporter, whereas the other is an unmodified peptide (hydrophilic) that is secreted by exocytosis. However, the biological significance of such asymmetric modifications of mating pheromones is not fully understood so far. Despite their different properties, the authors’ findings that a-factor and α-factor provide similar spatial information to guide polarity site movement in S. cerevisiae mating partner cells are interesting. In nature, however, a pheromone gradient may not be stably formed, meaning that polarization must be established differently from the mechanisms proposed for an experimental condition. Therefore, their significance should be interpreted more carefully.
In general, the writing is excellent. I have a few minor suggestions.
First, some studies on asymmetric features of mating pheromones in other yeasts, especially in the fission yeast Schizosaccharomyces pombe, are not fully cited. I encourage the authors to provide a fairer overview of the field in the introduction or discussion section. Second, the sentences (Lines 210-241) do not provide your data and findings. The Results section should minimize any information from other studies.
I cannot understand why β-estradiol was added to the medium.
Author Response
- Unstable gradients: We agree that in nature the gradients may not be stable. We do not think any of our interpretations rely on the assumption of stable gradients. Perhaps the reviewer is concerned about our comments on steady state gradients, but we note that given the rapid diffusion of pheromones, a steady state is reached quickly for any given pattern of secretion, which justifies discussion of steady state profiles.
- Fairer overview and citation of S. pombe data: we thank the reviewer for pointing this out, and have added comments on the interesting relevant papers: lines83-85 and 468-472.
- Removing introductory material from the Results section: we agree that the description of the findings from the Hartwell lab that motivated the experiments in Fig. 4 are not our "Results", but we believe that it is more effective to discuss that material in the Results section so that the motivation for our experiments is clear.
- Methods and use of β-estradiol: we apologize for the omission: new explanatory text has been added to make the protocol and methodology clear: lines 1121-127 and 138-145.
Reviewer 3 Report
Mating of budding yeast S. cerevisiae depends on the secretion of distinct mating pheromones, a and α factors. It is well established that the binding of these pheromones to cell-surface receptors triggers a signaling cascade, resulting cell cycle arrest in G1 phase, polarization of Cdc42, and the fusion of two haploid cells. Interestingly, a-factor is prenylated, but α factors is not. This research work directly tracks movement of the polarity site in each partner as a way to understand the spatial information conveyed by each pheromone. The results suggest that both pheromones convey very similar information. Some minor issues need to be addressed before publication.
- It would be better to include a diagram of the mating factor pathway in Figure 1. By doing so, it will be much easier for readers unfamiliar with this field to follow.
- More explanation about the angle of polarity site movement (θ) for Figure 1B is needed. Once the two polarity sites are close to each other, no significant θ increase is expected, but this is not reflected in panel B.
- This reviewer noticed that after kiss-and-run, the polarity site in the wild-type cells remained unchanged (Fig. 3A). It is not clear if this is the case for the WT cells in other pairs.
Author Response
- Thank you for the suggestion. A new cartoon has been added as Fig. 1A to explain the relevant aspects of yeast pheromones and receptors.
- Yes, we agree. More explanation has been added to the Figure 1 Legend and lines 172-173 in Methods.
- It seems to be often true, but we have not quantified that behavior.